# Photocathodic Protection of Cobalt Doped ZnO Nanorod Arrays for 316 Stainless Steel and Q235 Carbon Steel in 3.5 wt.% NaCl Solution

**Xiong Zhang** [1], **Guanghui Chen** [2], **Weihua Li** [3] **and Dianwu Wu** [1,*]

1   School of Materials Science and Engineering, Tongji University, No. 4800 Caoan Road, Shanghai 201804, China; zhangxiong@tongji.edu.cn
2   College of Chemical Engineering, Qingdao University of Science & Technology, No.53 Zhengzhou Rd, Qingdao 266042, China; guanghui@qust.edu.cn
3   School of Chemical Engineering and Technology, Sun Yat-Sen University Zhuhai Campus, Tangjiawan, Zhuhai 519082, China; liweihua3@mail.sysu.edu.cn
*   Correspondence: 1610446@tongji.edu.cn

**Abstract:** In this work, cobalt-doped ZnO nanorod arrays with anticorrosion function were successfully prepared on fluorine-doped tin oxide (FTO) substrates by a simple aqueous solution method. The XRD patterns and the energy-dispersive X-ray spectroscopy (EDX) analysis indicate the doped $Co^{2+}$ were successfully incorporated into the ZnO crystal lattice. The photocurrent density and open circuit potential (OCP) results indicate the photocathodic protection performance for 316 stainless steel (316 SS) and Q235 carbon steel in a 3.5 wt.% NaCl solution under a Xe lamp with a power of 300 W was enhanced with the increase of cobalt concentration, and the photoanode with a 15% Co/Zn ratio had the optimal photocathodic protection effect. The mechanism of enhancement may be result from the narrowed band gap, the lower recombination rate of the photogenerated electron-hole pairs, the intermediate impurity level, and the split of the hypo-outer shell of cobalt ions.

**Keywords:** cobalt; ZnO; photocathodic protection; corrosion protection

## 1. Introduction

Metal corrosion is a spontaneous process, in which the Gibbs free energy of the metal is lowered and the metal returns to its stable state in nature. However, corrosion causes great losses, including economic, environmental damages, life, injury, and efficiency. Among several corrosion control strategies, impressed current cathodic protection is not suitable for remote places, sacrificial anode protection consumes abundant resources every year, and the coating cannot tolerate few breakages, which may accelerate the rate of corrosion. Therefore, it is urgent for us to find a sustainable and environmentally friendly technology for corrosion control. Photocathodic protection has been studied extensively recent years. The principle is that electrons in n-type semiconductors are excited by solar light with certain wavelengths and transfer to a metal in a direct or indirect manner, which is equivalent to the impressed current cathodic protection, and cathodic polarization of the metal occurs. Photocathodic protection was originally found from the emergence of electron-hole pairs in a $TiO_2$ film when it was irradiated by ultraviolet light [1], and an open circuit potential (OCP) of the coupled metal shifted to a negative potential, which lowered and even stopped the corrosion of the coupled metal. The biggest advantage of photocathodic protection is that photocathodic protection is a green, sustainable and one of the most promising anticorrosion methods, which originate from the use of clean and inexhaustible solar energy for corrosion protection.

Up to now, most studies with regards to photocathodic protection systems focus on $TiO_2$ and its composites prepared by anodic oxidation with other semiconductors for anticorrosion of stainless steel (SS) [2–5]. Due to the low electron mobility, the photocurrent density of a pure $TiO_2$ coupled with 304 SS was smaller than 50 $\mu A/cm^2$, and the polarized OCP of the composite was approximately −450 mV. After the $TiO_2$/304 SS composite was modified with other semiconductors, its photocathodic protection properties were greatly improved. A 3D $ZnIn_2S_4$/$TiO_2$ composite increased the photocurrent density of the coupled 304 SS, which was greater than 2.0 $mA/cm^2$, and the polarized OCP of the $ZnIn_2S_4$/$TiO_2$ composite was negatively shifted to about −1.17 V. The photocurrent density of a $ZnS$–$Bi_2S_3$/$TiO_2$/$WO_3$ film coupled with 403 SS was greater than 110 $\mu A/cm^2$, and the OCP of the composite was about −520 mV. The study of $ZnIn_2S_4$/RGO/$TiO_2$ composites for the photocathodic protection of Q235 carbon steel (the rarely reported photocathodic protection for carbon steel) possessed a photocurrent density of 5.6 $mA/cm^2$ and an OCP of −1.1 V, which can provide adequate protection for Q235 carbon steel. However, most of the modified semiconductors are sulphides, which often suffer from the photocorrosion due to the instability of $S^{2-}$.

Zinc oxide (ZnO) is a semiconductor material with a wide band gap of about 3.2 eV, which are widespread used in photocatalysis [6], dye-sensitized solar cells [7], photoluminescence (PL) [8], and electronic devices [9], due to its abundance in natural resources, low price, and environmentally friendly features [10–12]. Besides, ZnO has high electron mobility, which is two orders of magnitude higher than that of $TiO_2$ [13]. ZnO might be an appropriate photoanode in photocathodic protection systems. It is well known that doping a selective element into ZnO is an effective route to improve optical and electrical properties, as well as increasing a carrier concentration, because a higher carrier concentration is required. Transition metal (TM) such as Co, Fe, Ni, and Mn doping into ZnO may increase the carrier concentration due to the impurity energy level or another introduced intermediate level [14], which may lower the requirement for a wavelength of light and promote the efficiency of photocathodic protection for metals.

In this paper, we successfully prepared ZnO nanorod arrays on FTO substrates with different cobalt doping concentrations by a simple aqueous solution method. The morphology, optical, and photocathodic protection properties of the as-prepared ZnO nanorod arrays were all changed with the change of cobalt concentrations.

## 2. Experimental

All of the chemical reagents used in this study were analytical reagents and without further purification. All of the aqueous solutions were prepared using double distilled water.

### 2.1. Preparation of ZnO Nanorod Arrays

ZnO nanorod arrays were prepared by an aqueous solution method. Firstly, FTO (surface resistivity ≤ 10 Ω/sq, size: 10 mm × 13 mm) substrates were ultrasonic cleaned with deionized water (DI water), acetone, ethanol, and DI water separately with a total time of 30 min. Then, these substrates were dried in a vacuum drying oven at 80 °C for 30 min. Secondly, in order to grow the ZnO nanorod arrays uniformly, the ZnO nanoparticles were coated on the FTO substrates. The well cleaned FTO substrates were spin-coated by a spin coater. A zinc nitrate hexahydrate ($Zn(NO_3)_2 \cdot 6H_2O$) and hexamethylenetetramine ($C_6H_{12}N_4$; HMT) aqueous solution was used as a precursor, and the spin parameter was set to third gears. The rotational speed is 500, 2000, and 5000 r/min, respectively, and the rotation time of each gear was 10 s. After that, the substrate was sintered in a muffle furnace at 450 °C for 10 min. This process was repeated for three times in order to form a uniform ZnO seed film on the substrate. Thirdly, the ZnO nanorod arrays were grown vertically by immersing the cleaned FTO substrate in the aqueous solution of $Zn(NO_3)_2 \cdot 6H_2O$ (0.025 M), HMT (0.025 M), and different concentrations of cobaltous acetate tetrahydrate ($Co(CH_3COO)_2 \cdot 4H_2O$). The reaction temperature and time were set at 80 °C and 12 h in the water bath, respectively. Then, the substrates were rinsed thoroughly with DI water for several times to eliminate residual salts, and they were dried for 1 h at

80 °C. On the basis of the different concentrations of $Co(CH_3COO)_2 \cdot 4H_2O$, the doping concentrations of cobalt in the reactive solution were 0, 1.25, 2.50, 3.75, and 5.00 mM, corresponding to the Co/Zn ratios of 0%, 5%, 10%, 15%, and 20%, respectively. The as-prepared ZnO samples with the five different doping concentrations of cobalt were labeled as C0, C1, C2, C3, and C4, respectively, in the subsequent discussion in this paper.

## 2.2. Characterization

The morphology and the crystalline structures of the cobalt-doped ZnO rod arrays were studied by field-emission transmission electron microscopy (FETEM, Tecnai G2 TF20, FEI, Hillsboro, OR, USA), scanning electron microscopy (SEM, EV018, ZEISS, Oberkochen, Germany), and X-ray diffraction (XRD, AXS D8 ADVANCE, Bruker, Madison, Germany) with a copper X-ray source (Cu-K$\alpha$, 50 kV, 250 mA). The analysis of the elements was conducted using energy-dispersive X-ray spectroscopy (EDX, A550I, IXRF, Austin, TX, USA) coupled with FETEM. The reflectance spectra were performed with an ultraviolet-visible-near infrared (UV-VIS-NIR) spectrophotometer (Cary 5000, Agilent, Palo Alto, CA, USA), and MgO powder was served as a reference material. The PL spectra were performed with a fluoro-spectrophotometer (F-4600, Hitachi, Tokyo, Japan). The measurements of the OCP and the photocurrent density of the photoanodes were performed on an electrochemical workstation (Reference 3000, Gamry, Philadelphia, PA, USA).

## 2.3. Electrode Fabrication and Photoelectrochemical Measurements

Metal electrode: Here, we chose widely used steels (316 SS and Q235) to be the objects of our research. First, these two cubic metals (length × width × thickness of 10 mm × 10 mm × 10 mm, the element contents of the two steels were shown in Table 1) were polished with sandpapers of 600, 1200, and 2000 meshes in sequence on a metallographic polishing machine. Second, to ensure good contact, the cubic metals were connected with a copper wire by tin welding. Third, one side of each cubic metal (to be studied) was remained exposed, and the other sides of the two cubic metals and their welded joints with ethoxyline resin were encapsulated to avoid corrosion. Before each examination, the exposed side of each cubic metal was polished to ensure the consistency and the veracity of the results.

**Table 1.** Element contents of 316 stainless steel (SS) and Q235.

| Type | C | Mn | Si | S | P | Ni | Cr | Mo |
|------|------|------|------|-------|-------|-------|-------|------|
| 316 SS | 0.08 | 1.80 | 0.90 | 0.029 | 0.045 | 14.00 | 17.00 | 2.00 |
| Q235 | 0.19 | 0.59 | 0.30 | 0.05 | 0.44 | – | – | – |

Photoanode: First, a ZnO/FTO substrate with a certain size (length × width of 10 mm × 13 mm, including a blank FTO with no ZnO films, of which the size was 10 mm × 3 mm) was cut out. Second, the conductive side of the FTO substrate was connected with the copper wire by silver conductive adhesive. Third, the blank FTO and the joint connected with ethoxyline resin were encapsulated.

Photoelectrochemical measurements: The diagrammatic sketch of the experiment setup is shown in Figure 1. The experiment setup was a double electrolytic cell system, which was composed of a photoelectric cell and a corrosion cell. The photoelectrodes and the metal electrodes (316 SS and Q235) were put into the photoelectric cell and the corrosion cell, respectively. The electrolytes in the photoelectric cell were $Na_2S$ (0.1 M) and $Na_2SO_3$ (0.1 M). The electrolyte in the corrosion cell was a simulated seawater solution (3.5 wt.% NaCl). The measurements of OCP and photocurrent density were conducted by a zero resistance ammeter (ZRA mode) on Gamry Reference 3000. The photoelectrode was connected with a working electrode (WE) and a working sense electrode (WSE). The metal electrode was connected with a current electrode (CE) and a current sense electrode (CSE), and a saturated calomel electrode (SCE) served as a reference electrode (RE). The measurements of OCP and photocurrent density were carried out under a simulated solar light source with a 300 W Xenon

lamp (Microsolar 300, Beijing Bofeilai Co., Beijing, China). All electrochemical measurements were performed at ambient temperature.

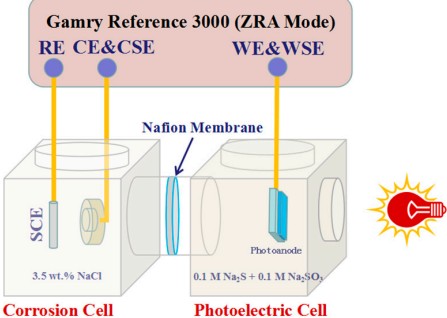

**Figure 1.** Schematic diagram of a double electrolytic cell setup for the measurements of open circuit potential (OCP) and photocurrent density of photoanodes.

## 3. Results and Discussion

### 3.1. Morphology and Crystal Structure Analysis

Figure 2 shows SEM images of the pure and cobalt-doped ZnO nanorod arrays grown on the FTO substrates. All the ZnO samples were regular nanorod-shaped, and the nanorods were all hexagonal prisms. It can be seen from the sectional-view SEM image of C0 (Figure 2a) that the one-dimensional ZnO nanorods vertically grew on the FTO substrate with a ZnO seed layer. The length of the pure ZnO nanorod was about 1.5 µm, and the thickness of the ZnO seed layer was about 200 nm. As shown in Figure 2b–f, the nanorod diameters of each sample varied, and the average diameter of the ZnO nanorods were about 80, 80, 80, 95, and 120 nm for C0, C1, C2, C3, and C4, respectively, which indicated the diameter of the ZnO nanorod became larger with the increase of the cobalt concentration in the precursor solution.

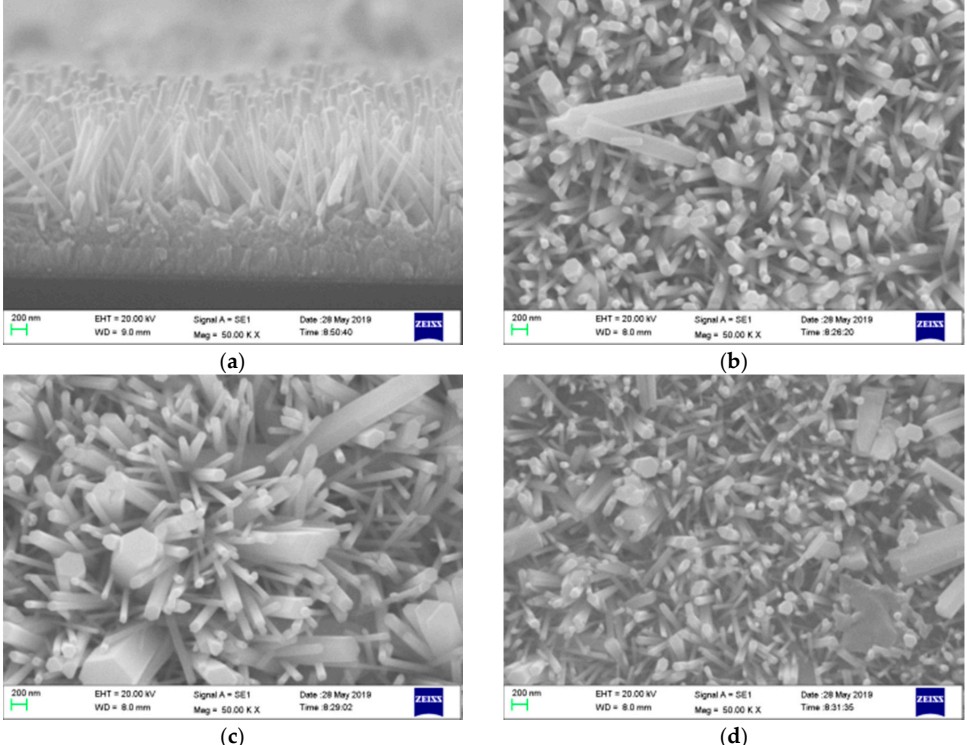

**Figure 2.** *Cont.*

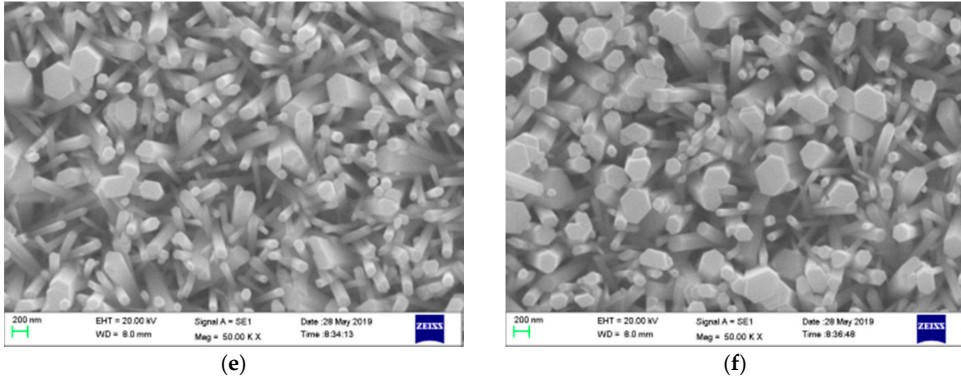

(e)                              (f)

**Figure 2.** SEM images of ZnO nanorod arrays grown on FTO substrates with different cobalt doping concentrations. (**a**) Sectional-view SEM image of ZnO nanorod arrays grown on an FTO substrate without cobalt doping, labeled as C0. (**b**) SEM image of ZnO nanorod arrays grown on an FTO substrate without cobalt doping, labeled as C0. (**c**) SEM image of ZnO nanorod arrays grown on an FTO substrate with a cobalt doping concentration of 1.25 mM, labeled as C1. (**d**) SEM image of ZnO nanorod arrays grown on an FTO substrate with a cobalt doping concentration of 2.5 mM, labeled as C2. (**e**) SEM image of ZnO nanorod arrays grown on an FTO substrate with a cobalt doping concentration of 3.75 mM, labeled as C3. (**f**) SEM image of ZnO nanorod arrays grown on an FTO substrate with a cobalt doping concentration of 5 mM, labeled as C4.

Figure 3a clearly shows that the as-prepared C3 sample was rod-like, and the diameter of the ZnO nanorod was about 100 nm. The FETEM image in Figure 3b shows a 2.684 Å (001) lattice fringe parallel to the basal plane, which provides confirmation that the rods were growing along the [001] direction. Figure 3c shows the EDX analysis of the C3 sample (the tested area marked with a red rectangular), and the result shows the existence of Co, Zn, and O in the spectrum. The existence of cobalt suggested that cobalt entered into the ZnO nanorod arrays. Table 2 shows the cobalt percentage in the doped ZnO rod arrays determined by EDX. There were 1.2%, 2.1% and 1.5% Co ions doped into ZnO rods in the C2, C3, and C4 samples, respectively. In addition, the actual quantities of cobalt in the C1 sample was too small to be quantificationally detected by EDX. The results implied that the actual quantity of cobalt in ZnO was far less than the quantity of the cobalt precursor, and the C3 sample had the maximum cobalt doping concentration.

In order to confirm that the dopants were doped into the crystal lattice of ZnO rather than generate the oxide of the dopant, XRD analysis was performed. Figure 3b displays the XRD patterns of the as-prepared pure and doped ZnO films. All the ZnO diffraction peaks are in good agreement with the JCPDS card (No. 36-1451) for a typical wurtzite-type ZnO crystal (hexagonal) [15]. These peaks at scattering angles ($2\theta$) of $32.06°$, $34.76°$, $36.56°$, and $47.84°$ corresponded to the diffractions from the (100), (002), (101), and (102) planes of the ZnO hexagonal phase, respectively. The (002) diffraction peaks of the cobalt-doped ZnO nanorod arrays, compared to that of the pure ZnO nanorod arrays, experienced slight shifts to the left by about $0°$, $0.2°$, $0.2°$, and $0.3°$ for C1, C2, C3, and C4 samples, respectively. The *c*-axis lattice parameter decreased with the increased concentration of the dopant incorporated into the crystal lattice. No other diffraction peaks were detected, implying that there were no $CoO$, $Co_2O_3$, or $Co_3O_4$ crystal structures in the doped samples.

**Table 2.** Cobalt concentrations in the ZnO nanorod arrays determined by energy-dispersive X-ray spectroscopy (EDX) analysis.

| Sample | C1 | C2 | C3 | C4 |
|---|---|---|---|---|
| Co concentration (%) | <1 | 1.2 | 2.1 | 1.5 |

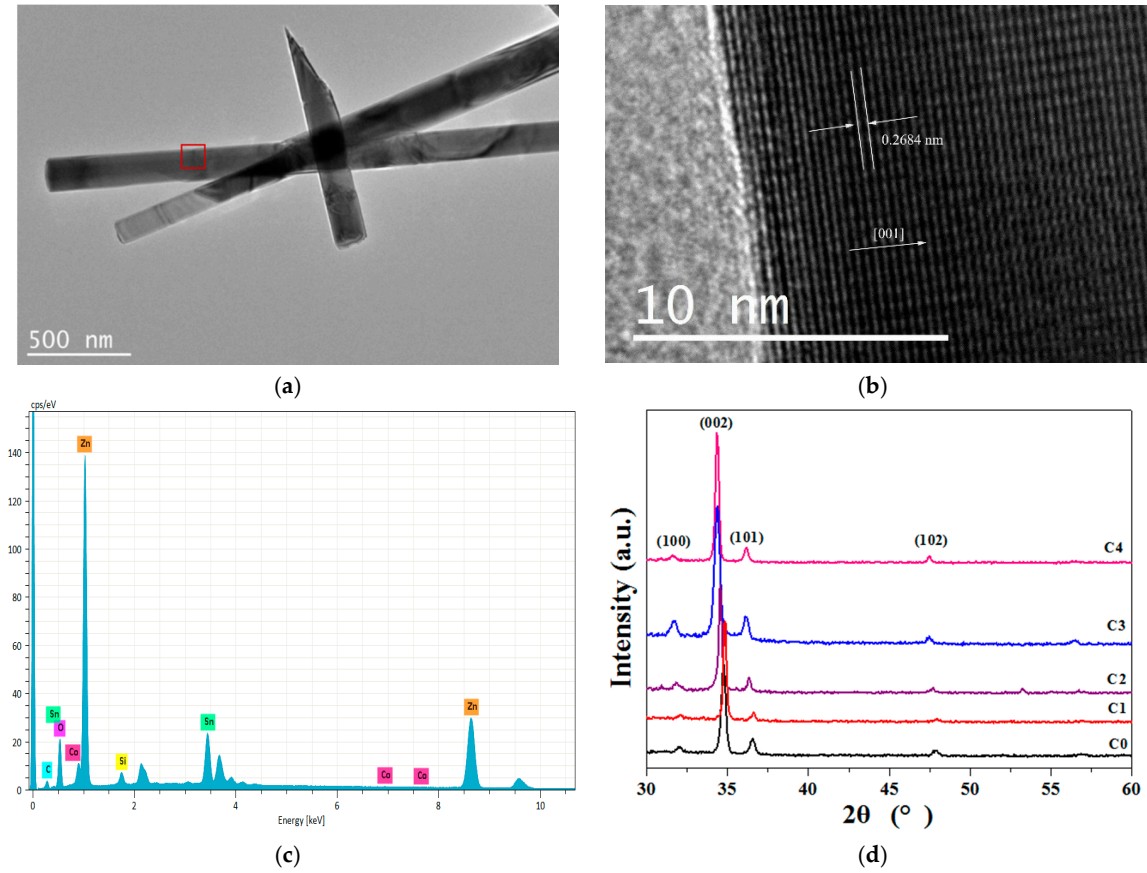

**Figure 3.** (**a**) TEM image of the C3 sample. (**b**) High resolution transmission electron microscope (HRTEM) image of the C3 sample. (**c**) EDX analysis of the C3 sample. (**d**) XRD patterns of the cobalt-doped ZnO nanorod arrays (C0–C4 samples).

### 3.2. Optical Properties Analysis

Figure 4a shows the UV-VIS diffuse reflectance spectra of the different cobalt-doped ZnO nanorod arrays. The absorption bandedge of the pure ZnO was at approximately 390 nm, and with the increase of the cobalt concentration, the absorption bandedges were slightly red-shifted. C3 had the largest red-shift. Compared to the relatively lower absorption intensities in the ultraviolet range, the cobalt-doped ZnO nanorod arrays had higher light absorption intensities in the visible light range (more than 400 nm), which meant the ability of utilizing visible light was improved.

The band gap of the photoanode could be roughly estimated by the following equation for a semiconductor [16]: $\alpha h\nu = A(h\nu - E_g)^\eta$, where $\alpha$, h, $\nu$, A, $E_g$, and $\eta$ represent the optical adsorption coefficient, the Planck constant, the frequency of light, a constant, the band gap of a semiconductor, and a characteristic of the type of electrons transition process ($\eta = 1/2$ for a direct semiconductor and $\eta = 2$ for an indirect semiconductor). As can be seen in Figure 4b, the estimated band gap values of C0, C1, C2, C3, and C4 films were approximately 3.23, 3.20, 3.19, 3.17, and 3.18 eV, respectively. The band gap became slightly narrowed, when the cobalt doping concentration in ZnO was increased.

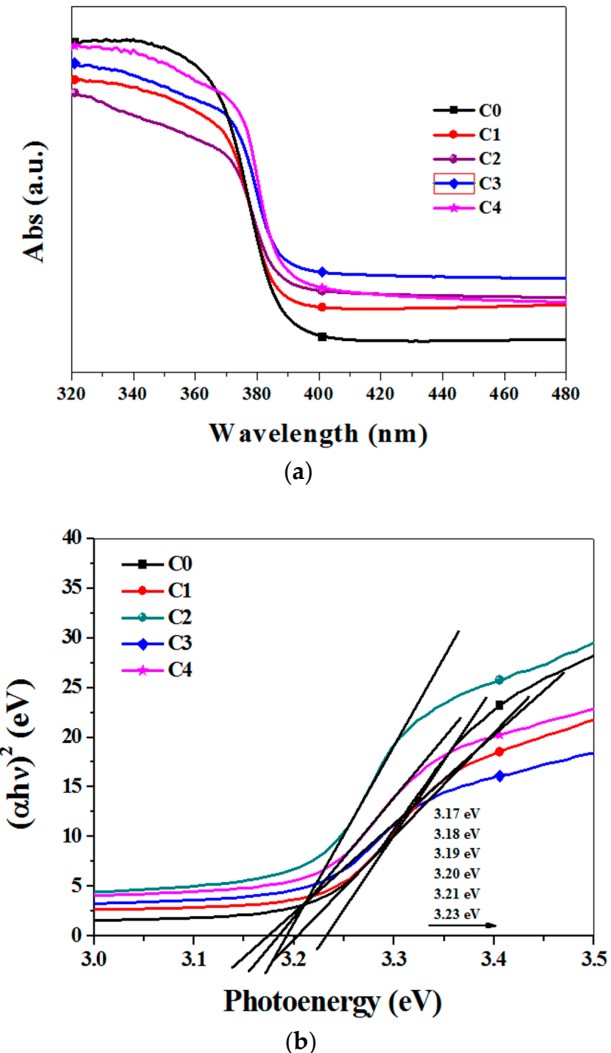

**Figure 4.** (**a**) UV-VIS absorption spectra of the pure and cobalt-doped ZnO nanorod arrays measured at room temperature. (**b**) Plots of $(\alpha h\nu)^2$ vs. photoenergy for the pure and cobalt-doped ZnO nanorod arrays.

Figure 5 displays the typical PL spectra of d the ZnO nanorod arrays with the different doping concentrations of cobalt. The PL spectra includes three peaks: a UV emission peak, which is located at 390–400 nm, a distinct shoulder peak located at ~460 nm, and an unobvious weak green emission band located at ~530 nm. The peak in the UV range belonged to the excitons recombination corresponding to the near-bandedge emission of ZnO [17]. The point of each photoanode's UV emission peak was indicated in the spectra with arrows, and it can be clearly observed that the UV emission peaks were red-shifted by 2, 8, 10, and 2 nm for C1, C2, C3, and C4, respectively, compared to that of the pure ZnO. This indicated that the band gap became a bit narrower with the increase of the doping Co concentrations before the Co/Zn ratio of 15% was reached, and the band gap turned larger, when the Co/Zn ratio was 20%. With the increase of Co concentration (Co/Zn ratio increased from 0% to 15%) in ZnO, the UV emission peaks obviously became weaker, which demonstrated that the near-bandedge emission caused by a recombination of photogenerated electron-hole pairs was weaker and a recombination of the charge carriers was depressed. The green emission peaks located at 530 nm have been reported to be ascribed to the internal defect in crystals (oxygen vacancies) and the transition of a photogenerated electron from a dark level below the conduction band to a deeply trapped hole [18]. We know that some dopants or defects in semiconductors tend to generate one or more deep energy

levels, which can not only capture electrons, but also capture holes, and these defects might be a recombination center in semiconductors. The recombination center can promote a recombination of the photogenerated electron-hole pairs and release energy in the form of fluorescence. As shown in Figure 5, with the increase of the Co doping concentrations, the defects in the photoanodes increased, and the corresponding intensities of the green emission peaks located at 530 nm was enhanced. It is not good for the utilization of the generated electrons. On the other hand, the recombination center between the valence band and the conduction band turned to be an intermediate energy level. The electrons on the valance band could be first excited to the intermediate energy level and then excited to the conduction band, leaving the holes in the valence band, although the energy between the valence band and the intermediate energy level or the energy between the intermediate energy level and the conduction band was significantly less than the $E_g$ of ZnO. The recombination center plays a role of a "step", and the electrons can be excited by a relative small energy through this "step". In other words, the electrons can be excited by light with shorter wavelengths, and it can be verified by UV-VIS absorption spectra (Figure 4) that the absorption intensities of the cobalt-doped ZnO nanorod arrays became stronger when the wavelength exceeded 400 nm.

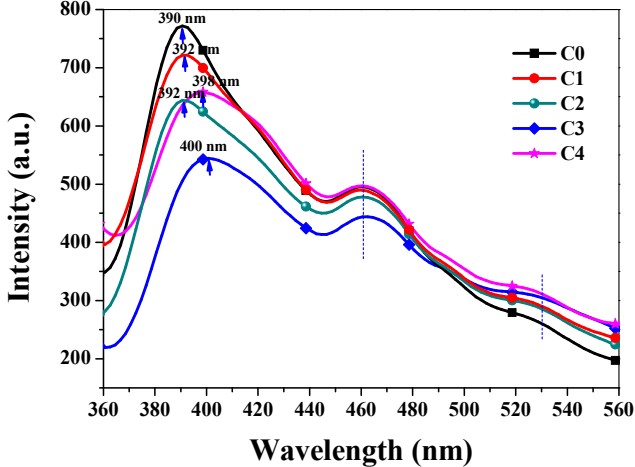

**Figure 5.** Photoluminescence (PL) spectra of the pure and cobalt-doped ZnO nanorod arrays.

### 3.3. Photocathodic Protection Properties

In order to validate the photocathodic protection properties of the doped ZnO samples, the OCPs and the photocurrent densities of the galvanic couples between the 316 SS electrode and different photoanodes under intermittent simulated sunlight in a 3.5 wt.% NaCl solution were measured. In the light of the principle of the cathodic protection for metal protection, the OCP of the coupled metal is quickly polarized with an impressed galvanic current, and the metal corrosion potential lowers the protective potential. As shown in Figure 6a, all the samples had relatively small photocurrent densities (lower than 10 μA/cm²) under dark conditions, and the photocurrent densities were all rapidly increased, owing to the photoelectric effect, once the light was switched on, which indicated that the photogenerated electrons originating from the photoanode flowed to the metal electrode. After the light was switched off, the photocurrents all returned to the initial position (dark conditions), and once the light was switched on again, the photocurrents were all rapidly increased again. After four circulations, the values of the photocurrent densities were as follows: C3 (136.27 μA/cm²) > C4 (110.41 μA/cm²) > C2 (97.88 μA/cm²) > C1 (68.23 μA/cm²) > C0 (42.69 μA/cm²). Compared with that of the pure ZnO, the photocurrents of the cobalt-doped ZnO samples all increased, and the photocurrent densities became larger with the increase of cobalt doping concentration. The value of OCP represents the thermodynamics trend of metal corrosion; the more negative potential the metal has, the smaller the probability that the metal will suffer from corrosion is. From Figure 6b, it can be seen that the corrosion potential ($E_{corr}$) of 316 SS in the 3.5 wt.% NaCl solution was approximately −164 mV, and all the OCPs

of the cobalt-doped ZnO samples became significantly decreased when coupling with the pure and cobalt-doped ZnO photoanode films. The OCPs sharply shifted to a negative potential immediately, once the light was switched on, which was originated from the polarization of 316 SS caused by the impressed current (photocurrent). The OCPs were unstable at first, and after manifold cycles of the light switched on and off, the OCP values became relatively stable, which may be attributed to the balance between the generation and the recombination of the photogenerated electron–hole pairs [19]. According to the theory of metal polarization, the larger the impressed current is, the stronger the cathodic polarization is, and the more negative potential the metal has. As a result of the values in Figure 6a, C3 had the most negative potential. Based on Figure 6b, the samples are shown in ascending order of the relative equilibrium potential: C3 (−955 mV vs. SCE) < C4 (−942 mV vs. SCE) < C2 (−848 mV vs. SCE) < C1 (−803 mV vs. SCE) < C0 (−579 mV vs. SCE). The OCP shifted to the original position, when the light was switched off, but was still lower than the $E_{corr}$ of 316 SS (lower than −240 mV vs. SCE). We can conclude that all the pure and doped ZnO photoanodes can provide effective protection for 316 SS, and the C3 film had the best protection property.

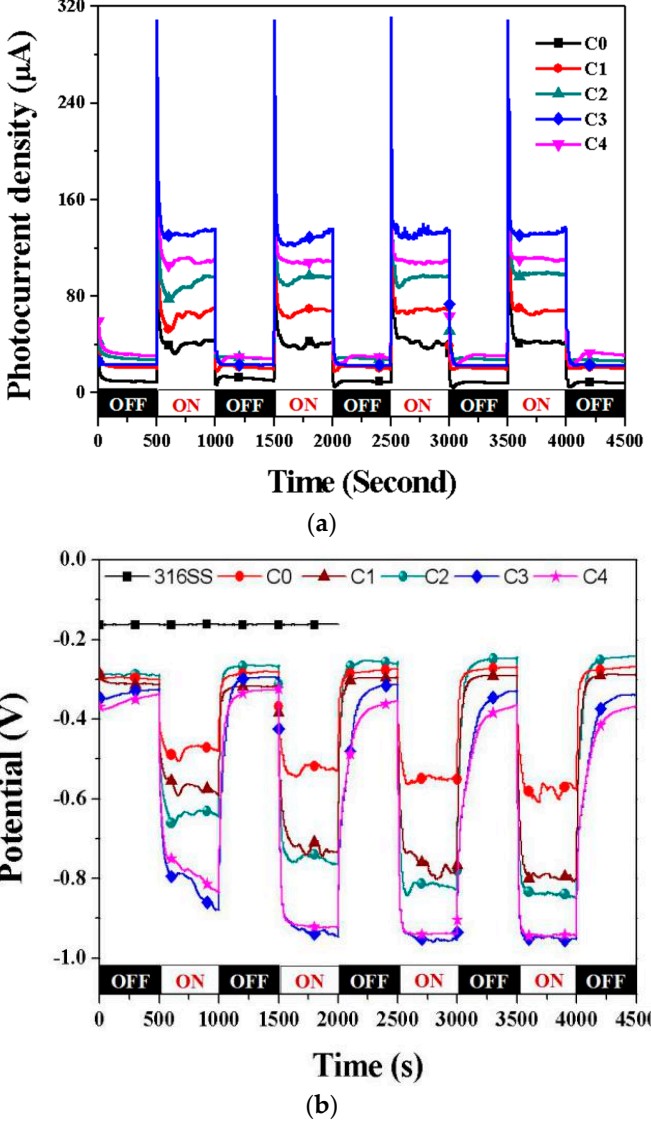

**Figure 6.** Photocurrent densities (**a**) and OCP (**b**) of the galvanic couples of the 316 SS electrode with ZnO doped with different cobalt concentrations under simulated solar light switched on and off intermittently in a 3.5 wt.% NaCl solution.

We also tested the photocathodic protection properties of Q235 with different ZnO photoanodes. The $E_{corr}$ of Q235 in the 3.5 wt.% NaCl solution (approximately −630 mV vs. SCE from Figure 7b) is more negative than 316 SS, so the corrosion protection for Q235 was more difficult, and greater demands were placed on the photoanode. Figure 7a shows the photocurrent density and OCP curves of the galvanic couples between Q235 electrode and different films under intermittent simulated sunlight in the 3.5 wt.% NaCl solution. Similar to that of 316 SS, the photocurrent density of the coupled photoanodes quickly became larger, once the light was switched on, and the order of the photogenerated current densities of the coupled photoanodes was shown as follows: C3 (133.02 μA/cm$^2$) > C4 (78.95 μA/cm$^2$) > C2 (65.84 μA/cm$^2$) > C1 (56.82 μA/cm$^2$) > C0 (36.48 μA/cm$^2$). When coupling with different photoanodes, the OCPs of all the ZnO photoanodes were higher than $E_{corr}$ of Q235, which indicates that all the ZnO photoanodes cannot provide sufficient cathodic protection for Q235 under dark conditions. When the light was switched on, the OCPs sharply shifted to the negative positions, which were all lower than $E_{corr}$ of Q235, and the photogenerated currents successfully gave rise to the polarization of Q235. Based on Figure 7b, the four photoandes are shown in ascending order of the OCP as follows: C3 (−1040 mV vs. SCE) < C3 (−1011 mV vs. SCE) < C2 (−994 mV vs. SCE) < C1 (−919 mV vs. SCE) < C0 (−888 mV vs. SCE). We can conclude that all the pure and doped ZnO photoanodes can provide effective protection for Q235 under irradiation.

Based on the tests of the photocathodic protection properties of the photoandes, it was demonstrated the pure and doped ZnO photoanodes had the ability of providing effective protection for 316 SS and Q235. In comparison with the pure TiO$_2$ [20,21], the SS and carbon steel coupled with ZnO had the more negative potentials, which might lead to the better protection effects on metals. In addition to the advantages of low price and abundance, the modified ZnO with better performance might be an appropriate photoanode for photocathodic protection.

### 3.4. Possible Photocathodic Protection Mechanism

In order to better understand the protection mechanism of the cobalt-doped ZnO nanorod arrays for 316 SS and Q235, we analyzed the reason in Figure 8 why the photocathodic protection efficiency was enhanced with the increase of cobalt concentration. Normally, electrons in the valence band are excited by the light and then transferred to the conduction band, leaving holes in the valence band. Some of the electrons are instantaneously neutralized with the photogenerated holes and release energy by ways of luminescence or heat. The other electrons migrate to the coupled metal, resulting in the polarization of the metal, and finally realize the protection of the metal. The holes in the valence band are consumed with the sacrificial agents (Na$_2$S + Na$_2$SO$_3$) in the photoelectric cell.

When cobalt is doped into ZnO, the band gap of ZnO firstly becomes slightly narrowed, and electrons can be excited by light with shorter wavelengths, which may increase the number of electrons that are excited from the valence band to the conduction band. Secondly, the doped cobalt in ZnO can form an impurity energy level between the conduction band and the valence band, and the electrons can indirectly be excited to the conduction band via this intermediate level, which greatly lowers the requirement for the energy of light. Thirdly, the hypo-outer shell of the TM in semiconductors is extremely prone to split [22]. In our experiment, Co ions were introduced in a ZnO crystal lattice, where Zn$^{2+}$ is located at the center of the tetrahedron built by four oxygen atoms. The hypo-outer shell of Co (*d* state) may be confined by the tetrahedral crystal field of ZnO and split into a higher triplet energy state and a lower doublet energy state. The triplet state is hybridized with the *p* orbital of the valence band soon afterwards and is split into two states: a bonding state in the valence band and an antiboding state close to the conduction band. The bonding state is localized and forms the Co–O band, and the antibonding state possesses a higher energy level and some itinerant electrons having higher probability of jumping to the conduction band. Based on the explanations above, with the increase of Co concentration, the recombination probability of photogenerated electron-hole pairs is reduced, and more electrons are possible to migrate to the conduction band to become free electrons. A large number of free electrons then flow to the surface of the metal, enabling the total metal to be in an electronic

surplus state, and the potential of the metal declines to a negative potential owing to the polarization. It is difficult for the metal atoms to lose electrons, resulting in the protection form corrosion.

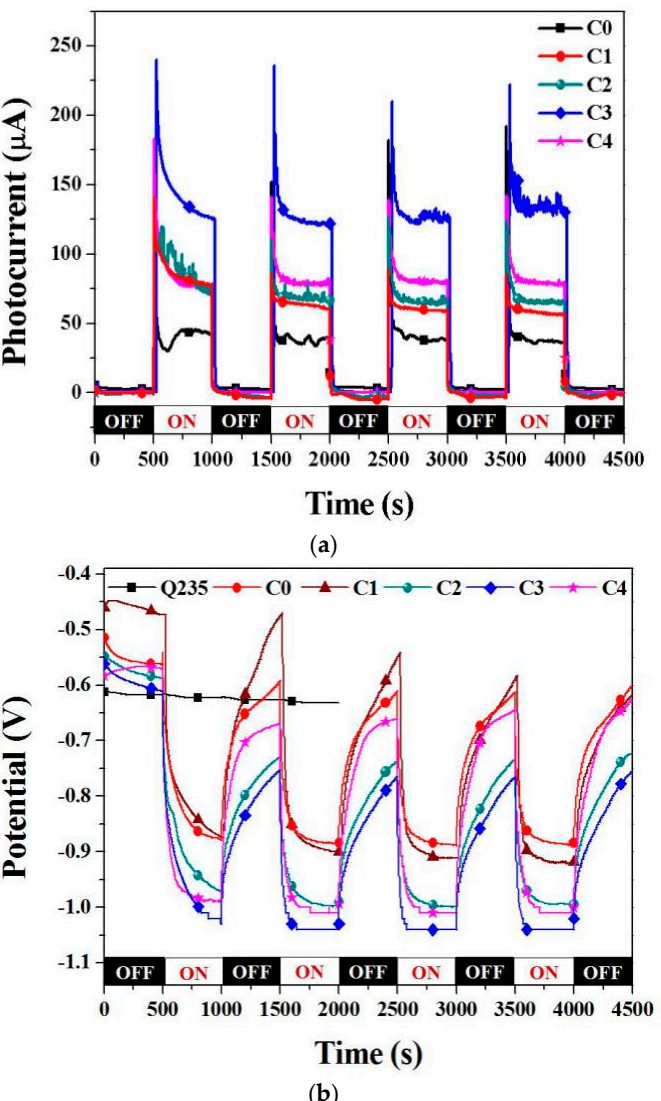

**Figure 7.** Photocurrent densities (**a**) and OCP (**b**) of the galvanic couples of Q235 electrode with ZnO doped with different cobalt concentrations, when simulated solar light was switched on and off intermittently in a 3.5% NaCl solution.

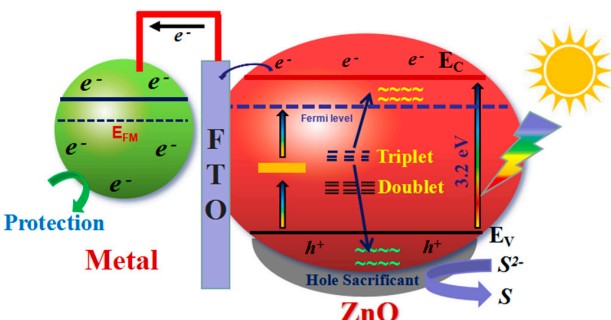

**Figure 8.** Schematic diagram of the anticorrosion process and the photogenerated charge transfer process in the doped ZnO coupled with metal.

## 4. Conclusions

Cobalt-doped ZnO nanorod arrays, which have good protection effects for metal, were successfully prepared by a simple aqueous solution method. The characterization results represent $Co^{2+}$ were successfully incorporated into the ZnO crystal lattice. The optical and photocathodic protection properties of cobalt-doped ZnO nanorod arrays were discussed, and particularly important features—UV-VIS absorption and PL spectra of the cobalt-doped ZnO nanorod arrays—were investigated, which illustrated that the band gap and the separation rate of the photogenerated electron-hole pairs of the ZnO nanorod arrays changed with the increasing the cobalt concentration in the ZnO nanorods. The OCP and the photocurrent density curves demonstrated that the photocathodic protection performance for 316 SS and Q235 in a 3.5 wt.% NaCl solution enhanced with the increase of cobalt concentration and the sample with a 15% Co/Zn ratio had the optimal photocathodic protection effect. These ZnO nanorod arrays have great potential for metal corrosion protection.

**Author Contributions:** Conceptualization, W.L. and D.W.; Investigation, G.C. and D.W.; Formal analysis, G.C. and D.W.; Project administration, X.Z.; Writing—original draft, D.W.; Writing—review and editing, D.W.; Funding acquisition, supervision, X.Z.

**Funding:** This work was financially supported by the National Key R&D Program of China (2016YFC0700800).

**Conflicts of Interest:** The authors declare no conflicts of interest.

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
