# Peer review of "Photocathodic Protection of Cobalt Doped ZnO Nanorod Arrays for 316 Stainless Steel and Q235 Carbon Steel in 3.5 wt.% NaCl Solution"

_coatings, doi:10.3390/coatings9120803_

Round 1

Reviewer 1 Report

Fot the manuscript with the title "Photocathodic protection of cobalt doped ZnO
3 nanorod arrays for 316 stainless steel and Q235 carbon
4 steel in 3.5 wt.% NaCl solution", i have some suggestions that are written below:   

1. The authors could provide the some STEM EDS analysis of the samples.

2. results and discussion section must be improved with examples from literature that sustain the authors suppositions.

Reviewer 2 Report

Well prepared work, however, I have some comments / questions:

1. Why was the NAFION membrane chosen?

2. On what basis were the electrolyte concentrations selected in properly prepared for different cells? - corrosion celll and photoelectric cell?

3. On what basis was chosen the ratio of Co:ZnO?

4. What roles did play Co and what ZnO in the system?

5. Figure 4 is not clear? please describe more...

6. Figure 8 should be modified

7. Have the authors read the review of ZnO dr Radzimskiej-Kolodziejczak?

I think they should, because there are many information about ZnO and aplication...

Round 2

Reviewer 1 Report

I accept the article in the present form.